

# Smartphone Pressure Data: Quality Control and Impact on Atmospheric Analysis

Rumeng Li[1], Qinghong Zhang[1*], Juanzhen Sun[2], Yun Chen[3],Lili Ding[4,5], Tian Wang[4]

[1] Department of Atmospheric and Oceanic Sciences, School of Physics, Peking University, Beijing 100871, China

[2] National Center for Atmospheric Science, Boulder, Colorado, United States

[3] National Meteorological Center, Chinese Meteorological Administration, Beijing 100080,China

[4] Moji Co., Ltd, Beijing, 100015, China

[5] Theme Tech Inc, Beijing, 100020, China

*Correspondence to*: Qinghong Zhang (qzhang@pku.edu.cn)

**Abstract.** Smartphones are increasingly being equipped with atmospheric measurement sensors, providing huge auxiliary resources for global observations. Although China has the highest number of cellphone users, there is little research on whether these measurements provide useful information for atmospheric research. Here, for the first time, we present the global spatial and temporal variation of smartphone pressure measurements collected in 2016 from the Moji Weather app. The data have an irregular spatiotemporal distribution, with a high density in urban areas, a maximum in summer and two

daily peaks corresponding to rush hours. With the dense dataset, we have developed a new bias correction method based on a machine learning approach without requiring users' personal information, which is shown to reduce the bias of pressure observation substantially. The potential application of the high-density smartphone data in cities is illustrated by a case study of a hailstorm occurred in Beijing in which high-resolution gridded pressure analysis is produced. It is shown that the dense smartphone pressure analysis during the storm can provide detailed information about fine-scale convective structure

and decrease errors from an analysis based on surface meteorological-station measurements. This study demonstrates the potential value of smartphone data and suggests some future research need for its use in atmospheric science.

## 1 Introduction

A lack of high-resolution observational data is one of the obstacles that limits the advance of numerical weather prediction (Bauer et al., 2015). This limitation can be extended to all areas in atmospheric research. In recent years, many new

observational technologies are emerging, including built-in smartphone sensors, such as those for pressure, temperature, humidity and aerosols (Overeem et al., 2013;Snik et al., 2014;Muller et al., 2015;Droste et al., 2017;Meier et al., 2017;Zheng et al., 2018). With over 2.7 billion people in possession of smartphones (Bankmycell, 2019) and an increasing trend of equipping smartphones with atmospheric measurement sensors, smartphone data can potentially be an auxiliary resource for global, high-density observations capable of resolving convective-scale features with a resolution lower than 2 km (Mass and

Madaus, 2014).




The smartphone sensors monitor atmospheric parameters and convert them into electrical signals, which can then be collected by different platforms, such as mobile weather applications. Low-cost smartphone sensor data have been used in several atmospheric research studies. Overeem et al. (2013) and Droste et al. (2017) used smartphone battery data to study air temperature and its application to urban heat islands. Snik et al. (2014) mapped atmospheric aerosols using smartphone
spectropolarimeters. Surface pressure is one of the most useful variables because it can reflect information about the whole atmospheric column and is less sensitive to the observational background (e.g., indoors/outdoors or the influence of the underlying surface vs. other variables like temperature and wind (Mass and Madaus, 2014;Hanson, 2016)); therefore, smartphone pressure data have received considerable attention from researchers. In addition to applications in weather forecasting (Mass and Madaus, 2014;Madaus and Mass, 2017;McNicholas and Mass, 2018b;Hintz et al., 2019), smartphone
pressure data can be used to monitor atmospheric tides (Price et al., 2018).

While smartphone pressure data may have potential value, they require validation and quality control before use. Price et al. (2018) and Hintz et al. (2019) showed that, although the variability between smartphone pressure data and meteorological-station observations is highly correlated, there exists noticeable bias. Price et al. (2018) calibrated the long-term stable bias using a one-point calibration method, while Hintz et al. (2019) developed screening methods to reduce observational noise.
Machine learning has also been applied to correct atmospheric pressure data (Kim et al., 2015;Kim et al., 2016;McNicholas and Mass, 2018a). Most previous publications on smartphone data calibration adopted a user-based approach, which required identification of each unique user and personal information. However, this raises privacy and ethical issues that pose a concern for the public. As highlighted by Muller et al. (2015) and Mooney et al. (2017), collecting as little personal information as possible and keeping raw data private are guiding principles of privacy preservation. Besides, without a stable
data collocation platform, performing user-based calibration can be time and resource consuming, especially for densely populated regions. It is therefore imperative to develop a new method that can efficiently calibrating smartphone pressure bias while protecting user privacy. It is worth noting that the need for such an effort has been recognized by other researchers and similar efforts are being undertaken (McNicholas, 2020).

China has one of the world's most densely distributed smartphone user bases (Bankmycell, 2019), which can potentially
produce highly dense observations. In this paper, we present, for the first time, a year-long dense and extensive smartphone dataset collected by the Moji Weather app, which is developed and operated by the internet environmental meteorological corporation Moji. The Moji Weather app is a popular smartphone weather app used in many countries, with a 53.90 % market share and more than 500 million users, as well as over 100 million weather queries made every day (Moji, 2019a, b). In the present study, we use the Moji smartphone pressure data for all of 2016 to show the spatial and temporal distribution
of the data set. With this highly dense network, we demonstrate the feasibility of a new machine learning bias-correction method that does not require users' private information, thereby ameliorating ethical issues. The dense network also makes it possible to study the detailed structure of atmospheric convection, which is demonstrated in this study by applying the bias corrected data to the fine-scale analysis of a hailstorm that occurred in Beijing.



This paper is organized as follows: Section 2 describes the data and methods used in our research. The statistical
characteristics of this dataset, bias correction results and its application to a hailstorm case are presented in section 3, 4 and 5
respectively. Conclusions and discussion are given in the final section.

## 2 Data and methods

### 2.1 Data description

Three types of data sets were used to perform this research: (1) Pressure data collected by a smartphone mobile weather
application every second in 2016. The application collects longitude, latitude, time and pressure data for each user, without
an unencrypted or encrypted ID. (2) Pressure data collected every five minutes by CMA (Chinese Meteorological
Administration) in 2016. There are 68,909 stations (including automatic weather stations (AWS) and conventional stations)
collecting meteorological data across the country, but only 13.32% of the stations make pressure observations. These
weather station surface data are used as the truth for bias correction of smartphone pressure data and for verification of
surface analysis. (3) Land-use and land-cover data for China in 2015 at a resolution of 1 km, which were accessed via the
Data Center for Resources and Environmental Sciences, Chinese Academy of Sciences (RESDC; http://www.resdc.cn).
These geographical data provide additional information necessary for our machine-learning-based bias correction method.

### 2.2 Quality control and preprocessing

The quality control and preprocessing procedure of the smartphone pressure data is described as follows. A workflow
diagram is shown in Fig. A1 summarizing the main processes of the procedure. First, a gross check is conducted: pressure
values lower than 890 hPa or higher than 1080 hPa are considered outliers and discarded (Kim et al., 2015;Madaus and Mass,
2017). The gross-checked data are referred to as GC-data hereafter. Next, we perform temporal and spatial averaging. As
described by McNicholas and Mass (2018a) and Hintz et al. (2019), there is a spin-up time for each measurement, and
location retrieval for smartphones has estimation error. To reduce such temporal and spatial errors, the GC-data are averaged
within a specified window of time and space. The time window size is 5 minutes to match the temporal interval of the
weather station data or 6 minutes to match the radar update interval whenever necessary. The spatial window size is $0.0001°$
latitude/longitude, i.e., the individual smartphone observation points are binned into specific sites with fixed locations to
eliminate the need for user IDs. The bias correction is then conducted on the aggregated data in a $0.0001° \times 0.0001°$ grid box
(~10m x 10m). In the rest of this paper, the aggregated data points will be referred to *smartphone sites* for convenience. The
next step in the quality control procedure is a neighbourhood check within each area of $0.01° \times 0.01°$ lat/lon. Data with
values greater than three times the standard deviation of the mean pressure in the area are removed. Finally, we perform a
statistical check. The boxplot approach is used to detect and handle climatological outliers (Iglewicz and Hoaglin, 1993). For
each boxplot, the upper quartile (Q3) is 75% for the smartphone air-pressure data and the lower quartile (Q1) is 25%. Data





that are 1.5 times the interquartile range (Q3-Q1) above Q3 and below Q1 are removed. The quality-controlled data after all
the above steps are referred to as QC-data hereafter.

It should be noted that the above quality control procedure does not include elevation correction of the pressure data not only
because the Moji smartphone data do not include the elevation information but also because the elevation-based pressure
correction may contain notable errors due to the uncertainties in GPS elevation positioning (Kaplan and Hegarty, 2006;Ye et
al., 2018) and in assumed pressure-height relations. As an alternative, we use a neighbourhood-based bias correction
approach, as described below, to correlate local pressure bias with land cover condition using the machine learning technique.

**2.3 Bias correction**

Previous studies have demonstrated the importance of implementing appropriate validation and bias correction procedures
before using smartphone pressure data in meteorological analysis (Muller et al., 2015;Hanson, 2016;McNicholas and Mass,
2018a). In our study, three machine learning techniques from the Waikato Environment for Knowledge Analysis (WEKA)
suite (Witten et al., 2011) are used to correct the smartphone pressure data and their effectiveness are compared.  Unlike
previous studies in which an individual model was trained for each smartphone, in this study, we developed a method,
named neighbourhood-based bias correction method, that trains a single model in a specified area rather than for a single
phone. Properly choosing the area size is crucial for the method to work effectively. It should be small enough to ensure
some degree of homogeneity in terms of geographical conditions and on the other hand, it cannot be too small because the
machine learning requires a large enough data amount to work properly. Since both users' behaviour and synoptic weather
background differ among seasons, we conducted the training for each season. The data were randomly separated into
training and test sets (Overeem et al., 2013). The parameters used as input in the machine learning are listed in Table 1,
including pressure from QC-data, longitude, latitude, time, land cover, number and standard deviation of raw-data
aggregated in a grid box, and distance of each *smartphone site* from the domain center. The land cover is used to provide
geographic information, which is an important input parameter for the neighbourhood-based bias correction approach. The
number and standard deviation of raw-data aggregated in a grid box are used to provide data uncertainty. The true pressure
value used for the machine leaning is provided by the 5 minutes pressure observations from AWS that are interpolated to
each *smartphone site*. To ensure some consistency of the two types of pressure data, training data with a pressure bias (the
difference of pressure values between smartphone and AWS) greater than 15 hPa are removed.
In order to evaluate the performance of the neighbourhood-based bias correction method, three experiments with the
following machine learning methods, multilayer perceptron (MP) (Pal and Mitra, 1992), support vector machine (SVM)
(Shevade et al., 2000;Smola and Schölkopf, 2004), and random forest (RF) (Breiman, 2001), were conducted and their
results will be compared later.



### 2.4 Objective analysis

It is well known that an accurate 2-dimensional surface analysis is extremely useful for nowcasting severe weather and
studying convective processes. Traditionally, this type of analysis is mainly obtained from surface weather station
observations. However, since most of the weather stations do not have pressure measurements, the surface pressure analysis
from them can only depict gross features of large-scale flow. The dense pressure observations from smartphones create an
opportunity to improve the surface pressure analysis. In this study we use an objective analysis method modified from

Barnes (1964) to conduct the analysis. The modified Barnes analysis method, described below, interpolates randomly
distributed data into a uniformly spaced coordinate system using a two-pass successive correction method.

If a variable $f_k$ is observed at a location $(x_{obs}, y_{obs})$, then the first pass analysis at a grid point $g_0(x_g, y_g)$ is obtained by Eq.
(1):

$$g_0(x_g, y_g) = \frac{\sum_k w_k f_k(x_{obs}, y_{obs})}{\sum_k w_k}$$  (1)

where the weight $w_k$ for the observation point is given by Eq. (2):

$$w_k = \begin{cases} \exp\left(-\frac{r_k^2}{\gamma L^2}\right), & r \leq r_e \\ 0, & r > r_e \end{cases}$$  (2)

where $r_k$ is the distance from the grid point $(x_g, y_g)$ to the k'th observation point; $\gamma$ is the convergence factor which controls
the refinement between the two passes (Barnes, 1973, 1974) and lies between 0 and 1 $(0 < \gamma \leq 1)$; L is the length scale that
controls the rate of fall-off of the weighting function; $r_e$ is the radius of influence within which the observations have impact

on the grid point. Different from the standard Barnes interpolation technique using a uniform length scale over the analysis
domain, an adaptive Barnes scheme is applied in this paper in which the length scale automatically adapts to data density, i.e.,
a spatially variable length scale is computed according to the data density.

The analysis in subsequent refinement pass is described by Eq. (3):

$$g_1(x_g, y_g) = g_0(x_g, y_g) + \frac{\sum_k w_k(f_k(x_{obs}, y_{obs}) - g_0(x_{obs}, y_{obs}))}{\sum_k w_k}$$  (3)

where $g_0(x_{obs}, y_{obs})$ is the estimate value of $g_0$ at an observation point which is given by bilinear interpolation.

The objective analysis method described above was applied to generate analysis fields with a 1km grid spacing for a
hailstorm case. In section 5, we will show that the high-resolution analysis fields can be used to analyze fine-scale pressure
patterns for the hailstorm.



## 3 Statistical characteristics

### 3.1 Spatial distribution

We used the GC-data to analyse the spatial and temporal distribution of the smartphone data counts in 2016. The data location map in Fig.1. shows that smartphone data are distributed over nearly all continents although most of the data counts occur in China with much higher data density (Fig. A2). The global mean density of the data is 40/bin/hour, whereas in China, the density is 176/bin/hour. The hourly pressure observation counts for the entire year of 2016 for China and its

surroundings (black box in Fig. 1.) are binned using a 0.1° × 0.1° grid and shown in Fig. 2a, which indicates that the data density is higher in megacities, such as the densely populated urban agglomerations of the Yangtze River Delta (Shanghai and nearby cities), Pearl River Delta (Guangzhou and nearby cities), and Beijing-Tianjin-Hebei region (marked by white circles in Fig. 2a). Because people carry mobile phones while traveling internationally, ship trajectories can be seen from two ports, the port of Shanghai (SH) and the port of Tianjin (TJ) (Fig. 2a.), but the amount of data at sea is much lower than

on land. However, in comparison with the Chinese Meteorological Administration (CMA)'s surface observations (Fig. 2b.), the amount and spatial coverage of the smartphone data are remarkable in nearly all regions.

### 3.2 Temporal distribution

The seasonal and diurnal distributions of the GC-data are displayed in Fig. 3. The data volume peaks during the northern hemisphere summer and reaches a minimum in winter (Fig. 3a, c). The annual mean data volume is 279,377/hour, which far

exceeds the value of 47,000/day in Korean shown in Kim et al. (2015), suggesting a large user base of the Moji Weather app. The data seasonality indicates that people check the weather more frequently in summer than in winter, owing to the fact that the app can only get the pressure information when users open it. The diurnal variation in global data volume (Fig. 3b, c) shows two peaks at 7:00 a.m. and 6:00 p.m. local standard time (LST), corresponding to the rush hour in the morning and evening respectively. Additionally, there is a steep decrease in data volume at night, consistent with a previous report that

smartphone data are inhomogeneously distributed throughout the day (Hintz et al., 2019). The diurnal distribution characteristic indicates that users tend to check the weather before going to work in the morning and getting off work in the evening. To demonstrate this more clearly, the spatial distribution of the standardized value of hourly data number at each site is computed for two days and displayed in Fig. A3. Interestingly, the data volume peak occurs earlier in northeast China, which corresponds well with an earlier sunrise (Fig. A3b.).

Analysis during a hailstorm occurred in Beijing further reveals that people respond promptly to severe weather event. The hailstorm occurred on 10 June, 2016 as a squall line passed through Beijing City from 1400 LST to 1700 LST. The hourly data volume on the day of the hailstorm and annual mean hourly data within 39°N–41°N, 115°E–118°E are plotted in Fig. 4. The diurnal cycle on the day of the hailstorm shows that, in addition to the two peaks at morning and evening, another peak appeared at 1600 LST with a data volume three times that of the annual mean. A 3D view of the data volume and radar echo

accumulated within 6 minutes (Fig. 4b-d) clearly shows a rise in data volume (Fig. 4b, d) as the storm approaches Beijing





and Tianjin and a drop after the storm passes (Fig. 4c.), which demonstrates the influence of severe weather on human behaviour

## 4 Evaluation of the bias correction method

Three neighbourhood-based bias correction experiments, each using one of the aforementioned machine learning methods, were conducted on a domain covering Beijing and its surrounding area from May to August, 2016. The machine learning bias correction was performed in each of the subdomains in Fig. 5c. using surface observations as the truth and the smartphone input parameters listed in Table 1. The region was affected by the 10 June, 2016, hailstorm and had a high density of smartphone pressure observations (Figs. 4a–d).

Constrained by the requirements of adequate data samples and reasonable computation cost, we chose 16 sub-domains of $0.25°$(longitude)$× 0.20°$ (latitude) in size. The pressure time series from two representative stations in Fig. 5a., b show that, although the trend of the weather station and smartphone is consistent, bias is clearly present, consistent with the results of Price et al. (2018) and Hintz et al. (2019).

Fig. 6. shows the mean absolute error (MAE) and computation time at different training regions for the three methods, it is evident that the RF method is more accurate and time-saving. The computation times for subdomain 7 and subdomain 8 using MP are more than 9 hours, so they are not shown in Fig. 6. From this comparison, we have found that the RF algorithm is more suitable for the neighbourhood-based bias correction of smartphone observations without requiring users' personal information. Furthermore, we discovered that the random data separation into training set and test set can cause random errors in the bias-corrected data, hence in order to eliminate these errors, the correction procedure was repeated for 50 times to generate an ensemble result.

Collecting smartphone data by weather app is convenient and common; however, the approach relies on the loyalty of users. Calibrating smartphone pressure individually can be only applied to data from long-lasting users, but it cannot be used for lately added users. In contrast, performing data correction for the aggregated data in a $0.0001° × 0.0001°$ grid box in a subdomain makes it possible for data from both user groups. In order to evaluate the applicability of our method on data from both types of users, we define the data sites appeared in both training set and test set as stable sites and those only appeared in test set as additional sites. To quantify the performance of bias correction, the domain average MAE and standard deviation of ensemble mean for the 16 subdomains are displayed in Fig. 7 for the raw and bias-corrected data from both the stable sites and the additional sites. The MAE was calculated using data from the *smartphone sites* for each subdomain. Comparing the MAEs between the raw (Fig. 7a) and bias-corrected data (Fig. 7c), it is evident that the neighbourhood-based bias correction method is capable of substantially reducing the MAE not only for the stable sites but also for the additional sites with slightly more reduction for the stable sites (from 5.95 hPa to 0.53 hPa) than for the additional sites (from 5.90 hPa to 0.99 hPa). It is also shown that the method reduces the MAE spread by 78% for the stable sites and by 16% for the additional sites (Fig. 7b., d). The less MAE and spread reduction for the additional sites is not



surprising because they are newly added data with shorter data history, and hence have less data samples (Fig. 8a, b).
Encouragingly, our results suggest that the neighbourhood-based method can partially mitigate the difficulty related to
recently added data with shorter data history. In comparison with the bias correction method based on single site, the
neighbourhood-based method resulted in a MAE substantially smaller (see Fig. 8c, d).

## 5 Impact of smartphone data on hailstorm analysis

High density pressure observations can potentially help identify small-scale surface pressure patterns beneath a thunderstorm
(Johnson and Hamilton, 1988). Although the quality-controlled gridded smartphone pressure data reduce the number of data
points, they are still adequate to represent the fine-scale pressure patterns. In this section we first show what small-scale
information the quality controlled high-density pressure data at the *smartphone sites* (with a spatial resolution of 0.0001°, or
approximately 10 m) can provide and then demonstrate the impact of the smartphone data on gridded 1-km pressure analysis
that are obtained using the objective analysis method described in section 2.4.

Fig. 9. shows a composite plot of radar reflectivity, pressure changes calculated from surface weather station observations
and from smartphone data, as well as wind and equivalent potential temperature from the station observations. To be
consistent with the time interval of radar volume scan, the smartphone QC-data averaged every 6 minutes were used to
generate the 6-minute pressure tendency. Further, because the weather station data are at a 5-minute interval, the pressure
change and temperature from these data are shown at times closest to those of radar volume scan. Since there are only 15
weather stations providing pressure observations in this region, they are unable to locate the leading edge of the cold pool. In
contrast, the smartphone pressure observations are much denser and hence is able to capture the fine-scale pressure change
associated with the cold pool, as depicted in Fig. 9. by the "×" symbol representing the 6-min change of perturbation
pressure (i.e., domain mean subtracted) greater than 0.52 mb. Compared with the cold pool leading edge identified by $\theta_e$,
following Schlemmer and Hohenegger (2014), from the analysis of surface observations, the leading edge of the cold pool
based on the smartphone pressure change is about 10 km ahead at 1506 LST (Fig. 9b.) and quite close at 1524 LST (Fig. 9c.).
At 1454 LST (Fig. 9a.), the pressure change is largely negative ahead of the cold pool whereas Fig. 9d. mainly shows
negative pressure changes after the leading edge has passed the area; both are consistent with the surface station observations
but more detailed.

We conducted three objective analysis experiments using the method described in section 2.4 to demonstrate the potential
benefit of using smartphone observations along with surface weather station observations to improving surface pressure
analysis, i.e., the experiment SFC using only weather station pressure observations, SP using only smartphone data, and
SFC+SP using both the station and smartphone data. The analysis grid spacing is 1 km. Fig. A4. shows the domain for
surface analysis, the locations of Beijing Radar and the surface stations.

The analyses of perturbation pressure (i.e., relative to domain mean) from the experiments SFC (Fig. 10a, c, e) and SFC+SP
(Fig. 10b, d, f) are compared at 1500, 1506, 1512 LST in Fig. 10. To illustrate the coupling between pressure and wind in the


storm region, the wind field at 150 m from VDRAS (Variational Doppler Radar Analysis System) and the composite reflectivity observation are overlaid. VDRAS is a rapid update analysis system based on the variational technique that blends radar radial velocity and surface wind observations to produce 3-dimensional wind analysis (Sun and Crook, 1997, 1998). We first note that the perturbation pressure analysis from SFC+SP (right column) displays small-scale features in and around the storm that are absent in SFC (left column). The high center of pressure perturbation is nearly collocated with the center

of the outflow near the northwest flank of the main body of the storm system (Fig. 10b, d, f). The vertical cross sections shown in Fig. 11 through the line A-B (see Fig. 10) indicate that the high-pressure perturbation corresponds to the rear-flank downdraft aloft behind the intense radar echoes of the southeastward moving convective system. Although the relatively low-pressure regions are seen in front of the convective system in both experiments but only the SFC+SP experiment captures the relatively low-pressure region northwest of the system. The overall distribution pattern of pressure perturbation

in SFC+SP is consistent with the conceptual model of Markowski and Richardson (2010), but the current analysis reveals that the surface high pressure region and low level divergence center slightly lag behind the center of the intense reflectivity echoes rather than right beneath it as in their conceptual model. We believe the difference is resulted from the higher resolution of the smartphone data applied in this study, but further studies are needed to draw a definite conclusion. Furthermore, the pressure analysis from SFC+SP provides more detailed information about storm evolution than what is

shown in SFC. As the storm moves southeastward, the cell in southwest, denoted as cell 2 in Fig. 10., separates into two (Fig. 10b) and the northern one merged into cell 1 (Fig. 10d, f). During the merging process, the high-pressure region behind cell 1 becomes stronger and wider, which may indicate the enhancement of cell 1 in correspondence with the increased downdraft and updraft as shown in Fig. 11c.

Analysis accuracy for the two experiments was verified against the 15 weather station pressure measurements in the domain.

In order to avoid dependence between the analysis and verification, both experiments were repeated 15 times; each alternately excludes the measurement from the specific station to be verified against. The temporal distributions of MAE between model analysis and observation at different surface stations are shown in Fig. 12. The results confirm that the experiment SFC+SP reduces the analysis error at most stations, even at those around which there are relatively less smartphone observations, such as the stations XH and LF. Although at the stations where there are much fewer smartphone

observations, such as GA and ZZ, the analysis with smartphone pressure data alone in the experiment SP results in larger error than in the experiment SFC, adding the station observations in SFC+SP results in reduced analysis error (Fig. 12 n, o). The correlation between the smartphone data density and the analysis accuracy is more clearly illustrated by Fig. A5., which shows that the MAE is less than 0.20 hPa as long as there are more than three *smartphone sites* around the verifying weather station measurement.

In summary, our quantitative verification results demonstrate that the high-resolution smartphone data generally improve surface pressure analysis in comparison with the weather station data, combining these two datasets results in further improvement especially at the locations where the smartphone data are sparse.



## 6 Conclusions and discussion

This study focused on smartphone pressure data acquired from the Moji Weather app in 2016 and showed their
characteristics for the first time. A neighbourhood-based bias correction method applying machine learning techniques was
developed without any privacy information needed. The bias-corrected data were employed to explore the potential value of
these data for improving atmospheric analysis through a case of a hailstorm in Beijing, China.

Since these data are produced by citizens at large, their spatial and temporal distributions are affected by human behaviours.
It was shown that the data are mostly distributed around urban areas, and data volume peaks during summer. There is also a
diurnal cycle in which the data volume is higher during the day than at night, with two peaks appearing at 0700 LST and
1800 LST. Our case study showed an anomalous increase in data volume when the hailstorm occurred, suggesting that
public concern increases in anticipation of high-impact weather situations, which means the data can be useful for disaster
prevention.

We proposed and demonstrated a neighbourhood-based bias-correction method that can address user privacy issues. Despite
growing concern from the public regarding personal privacy, little studies have addressed how to circumvent the problem.
Since Moji protects data privacy during the collection and processing stages, no private information was included in the raw
data that we received; and the bias correction method proposed in this study does not require such information. Our results
showed that the MAE and MAE spread can be successfully reduced not only for long-term stable sites but also for lately
added sites that present a challenge using the traditional user-based bias correction method.

With this feasible and effective bias correction method, the potential utility of the high-resolution smartphone data
(approximately 10 m horizontal resolution) is shown using a hailstorm case. We have found that the 6-min pressure change
can provide convective-scale information such as cold pool leading edge, especially in megacities, where the data are most
dense. Using a modified Barnes objective analysis method on a 1-km grid, we also showed that the data can be used in
conjunction with weather station data to improve surface pressure analysis. The analysis is capable of depicting the high
pressure associated with the rear-flank downdraft of the hailstorm and temporal variation of pressure perturbation related to
the splitting and merging process within the convective system.

Through the current study, we have gained an understanding of the smartphone pressure data characteristics, developed a
practical and effective quality-control and bias correction method, and demonstrated the value of the data in surface
objective analysis, our next step is to explore whether the data can be useful in improving convective weather forecasting
through data assimilation. Previous data assimilation research with smartphone pressure data mainly focused on assessing
whether the data have a positive impact on regions where weather stations are not available (McNicholas and Mass,
2018b;Hintz et al., 2019). However, it may present greater challenge to demonstrate that the smartphone data can yield
additional benefit to the existing weather station network mainly because of the uneven distribution of the smartphone data
across the globe. Efforts are needed to develop data assimilation approaches that can make best use of the smartphone data in
numerical weather prediction models by taking into account the characteristics of these data. The current study also points to



the need of an improved smartphone data collection mechanism. The data volume collected by a weather app relies heavily on the popularity of the application that serve as the data-collection platform (Kim et al., 2015;Hintz et al., 2019). As such, the data distribution relies heavily on the severity of local weather. Thus, a more stable and widely used platform is needed to provide useful high-resolution global observations without a correlation to local weather. Additionally, the smartphone

information included in our research is limited, additional auxiliary information, such as smartphone models, sensor types, and the altitude at which smartphone data were measured, would be conducive to the bias-correction procedure and subsequent analysis.

**Data availability**

The land-use and land-cover data is available on the website http://www.resdc.cn. Smartphone data, surface observation and

radar data are provided by Moji Corporation and the Chinese Meteorological Administration, and are available on demand.

**Author contributions**

The analysis and figures were produced by RL, QZ and JS contributed to the data analysis and supervised the writing of and revised the manuscript. YC, LD, TW provided the data quality control method used in the manuscript.

**Competing interests**

The authors declare that they have no conflict of interest.

**Acknowledgements**

This study was supported by the National Natural Science Foundation of China Grant No. 41875052.

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





**Table 1. Parameters used for machine learning.**

| Data type/source | Field | Description |
|---|---|---|
| Smartphone | Gridded pressure at each *smartphone site* | Pressure to be corrected |
| | Longitude | Location information |
| | Latitude | Location information |
| | Time | Time information |
| | Land cover | Geographical information |
| | Number of pressure observations aggregated at each *smartphone site* | Data Uncertainty |
| | Standard deviation of pressure observations at each *smartphone site* | Data Uncertainty |
| | Distance from domain center | Additional location information |
| Automatic Weather Station | Pressure observation interpolated to each *smartphone site* | "True" pressure |

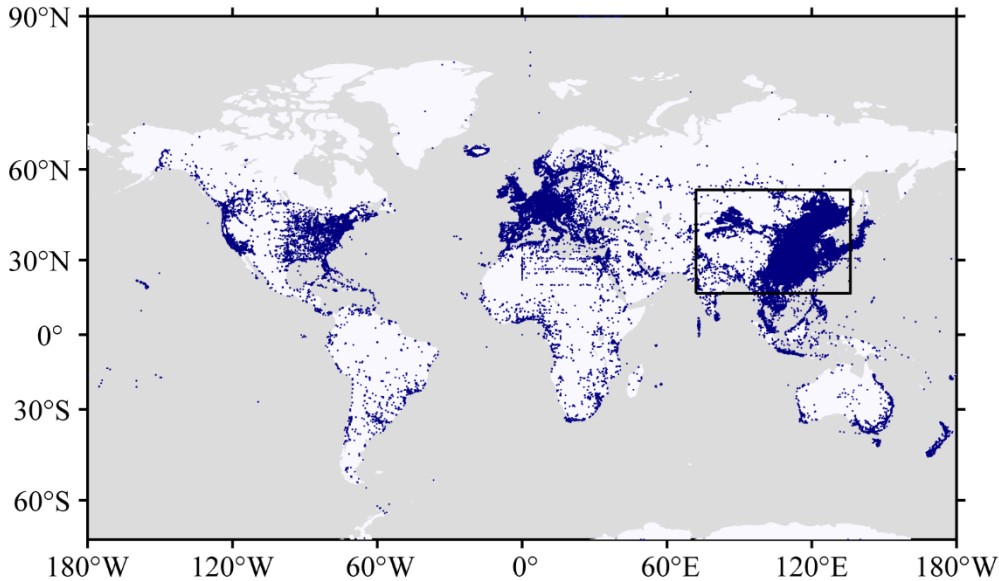

**Figure 1. Locations of global pressure observations in 2016 from the Moji Weather application.**

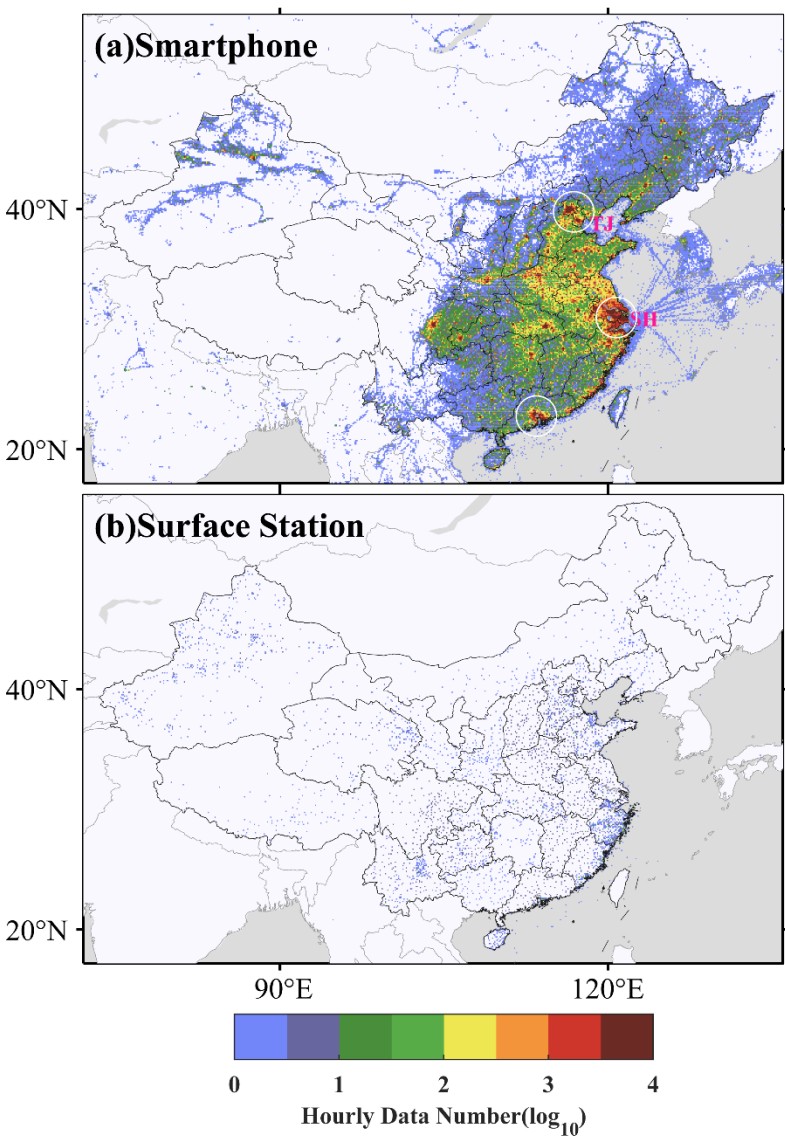


**Figure 2. (a) Hourly pressure observation counts (log10 transformed) averaged over the year 2016. (b) Same as a but for the China Meteorological Administration (CMA) surface stations. Data are binned into a 0.1° × 0.1° grid in (a) - (b). The location of the port of Shanghai and the port of Tianjin are labelled as "SH" and "TJ" in (a). The white circles indicate the urban agglomerations of (from north to south) Beijing-Tianjin-Hebei region, Shanghai and nearby cities, and Guangzhou and nearby cities.**





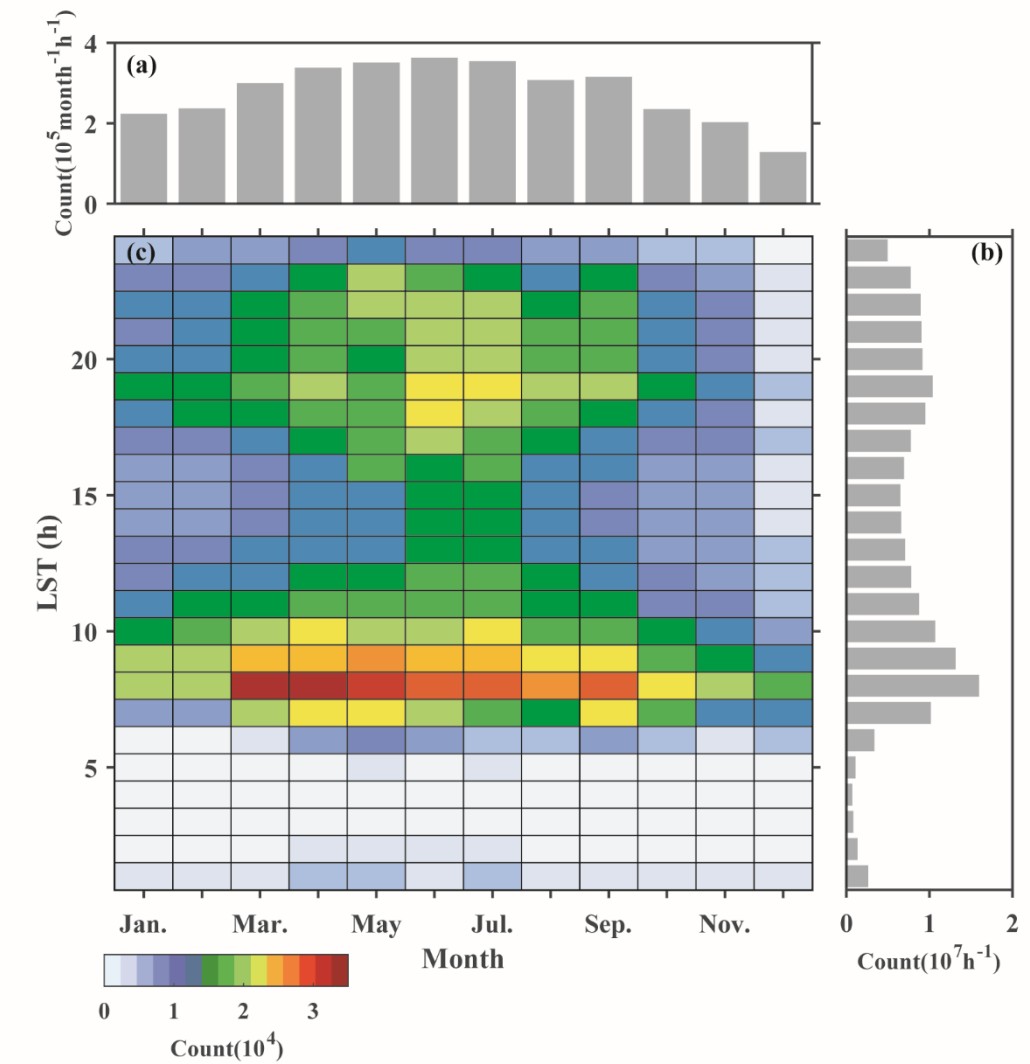

**Figure 3. (a) Seasonal variation of global hourly counts of smartphone data for each month. (b) Diurnal variation of global smartphone data counts. (c) Annual mean hourly data count at different local standard times (LST) and months.**



**Figure 4. Diurnal variation of the data volume for smartphone data on the day of the hailstorm (red line) and the annual mean**
**value (blue line) for 39°N–41°N, 115°E–118°E. (a)–(d) show a 3D view of data counts in a 0.05° × 0.05° grid over 6 minutes (colored**
**columns) before each radar volume and radar echo (grey columns). The color and height of each column represent the value of**
**the data count. BJ, Beijing; TJ, Tianjin.**



**Figure 5. (a, b)** Pressure time series during the training period for the AWS (black line) and smartphones (red dots); smartphone
pressure was interpolated into station location using the inverse distance weighting method. **(c)** The domains of the machine
learning area. Shaded: regional land use (PF, paddy field; RC, rainfed cropland; CFL, closed forest land; S, shrubbery; SWL,
sparse woodland; OWL, open woodlot; HCG, high-coverage grassland; MCG, moderate coverage grassland; LCG, low-coverage
grassland; G, graff; L, lake; R, reservoir pond; PGS, permanent glacier snow; TF, tidal flat; FL, flood land; UL, urban land; RSA,
rural settlement area; OCL, other construction land; S, sand; Go, Gobi; SAL, saline-alkali land; W, wetland; BE, barren earth;
BER, bare exposed rock). Automatic weather stations (AWS) with pressure observations are shown by black triangles.

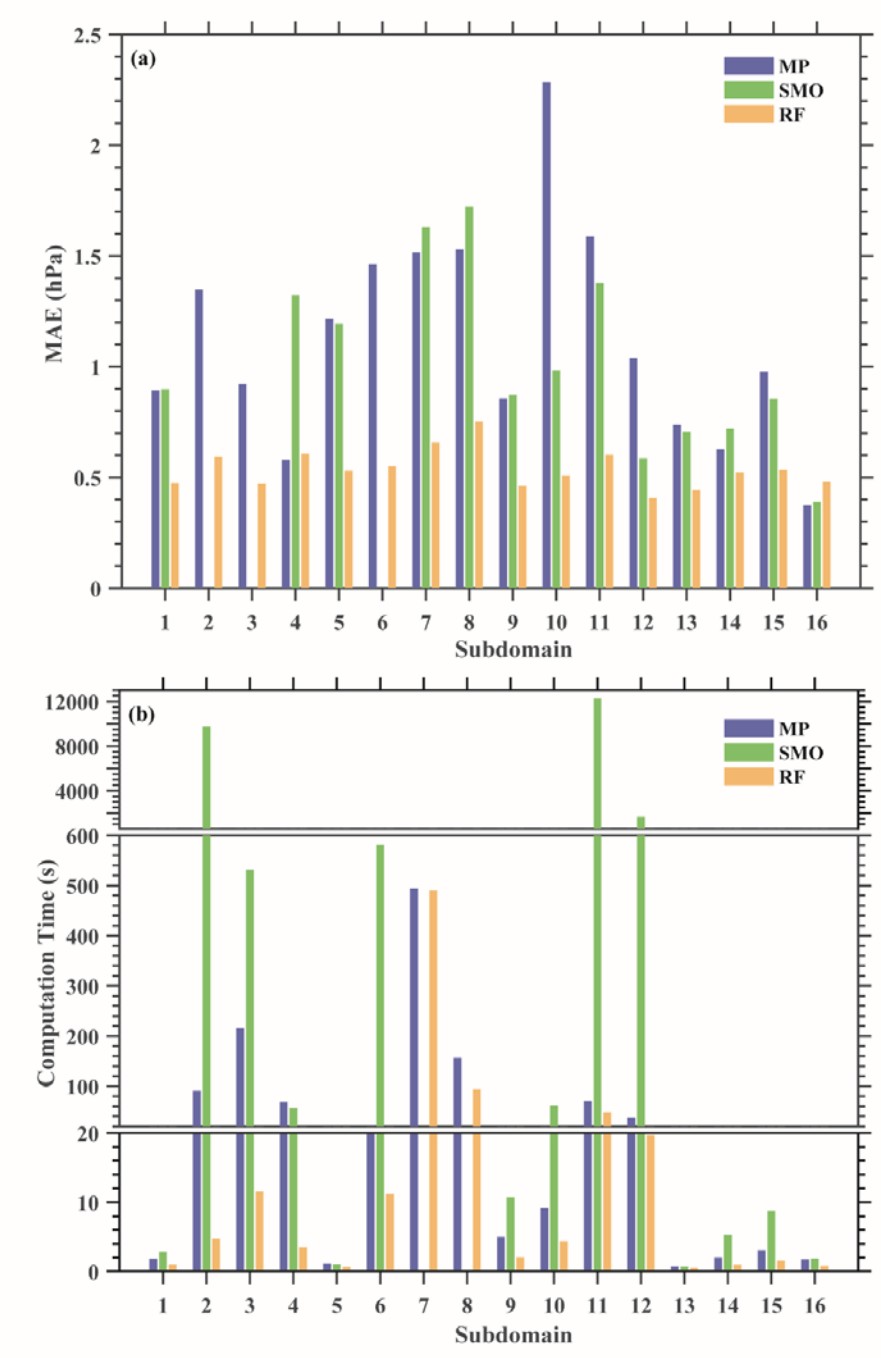

**Figure 6. (a) Mean absolute error (MAE) distribution at different training subdomains for different machine learning methods (MP refers to Multilayer Perception method; SMO refers to Support Vector Machine method; RF refers to Random Forest method). (b) is the same as (a), but for computation time.**

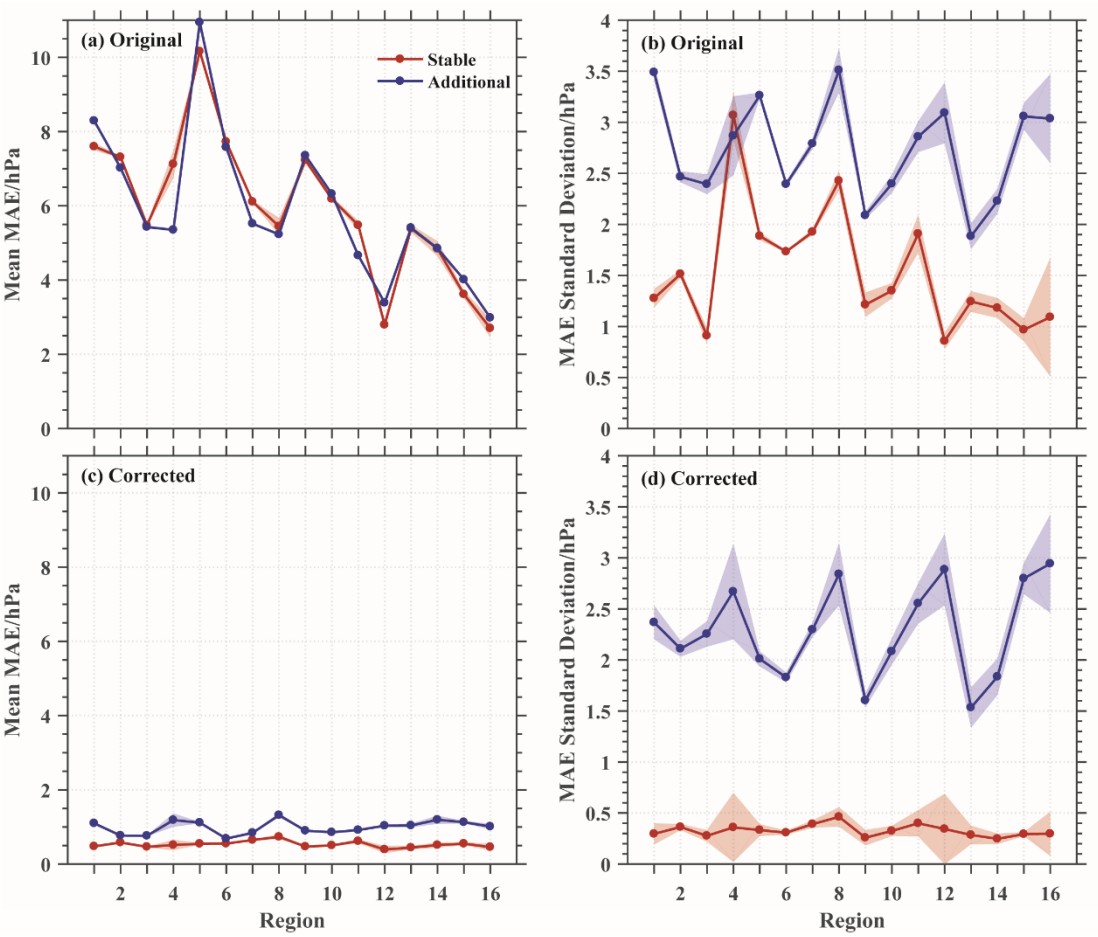

**Figure 7. Distribution of domain average mean absolute error (MAE; the left panel) of ensemble mean and standard deviation (the right panel) for different subdomains for the original dataset (a, b) and the bias-corrected dataset (c, d). The line marked by dots is the ensemble mean value and shading is the double standard deviation ensemble spread. Red line is the result for stable sites and blue line is the result for additional sites. See text for the definitions of stable sites and additional sites.**



**Figure 8. (a, b)** Scatter plot of mean absolute error (MAE) versus data number of observation sites (a) for stable sites, (b) for additional sites; **(c, d)** Scatter plot of station pressure versus bias-corrected pressure for (c) the neighbourhood-based method and **(b)** single site method of one ensemble member. Averaged mean absolute error (MAE) of two methods are shown in the plot.

**Figure 9. The 5-minute pressure change from surface stations (triangles) and the 6-minute pressure change from smartphones (points), temperature observations from surface station (green contour), wind field (black arrow) and composite radar reflectivity (shaded) during a hailstorm that occurred on 10 June, 2016, in Beijing, China. The station pressure change is shown at the time closest to that of the radar volume. The "✕" symbol marks the locations where the 6-minute pressure change perturbation is greater than 0.52 hPa. The blue dashed line is the $\Delta\theta_e = -6$ K isoline from analysis of surface observations ($\Delta\theta_e$ is defined as the difference between the equivalent potential temperature at a point and the domain averaged value).**

**Figure 10. Surface pressure perturbation analyses (shaded) from the experiment SFC (a, c, e) and SFC+SP (b, d, f), overlaid by VDRAS wind field at 150 m (thick black arrows) and column maximum radar reflectivity (contours). The valid time is 1500 LST for the top row, 1506 LST for the middle row, and 1512 LST for the bottom row.**





**Figure 11. Vertical cross section of the radar reflectivity (shaded), VDRAS wind field (thin black arrows) and vertical velocity field (brown contours with dash lines for downward motion and solid lines for upward motion) along the line A-B in the Figure 10. The solid blue line and red dash line are the surface pressure perturbation along the A-B line from SFC and SFC+SP, respectively.**



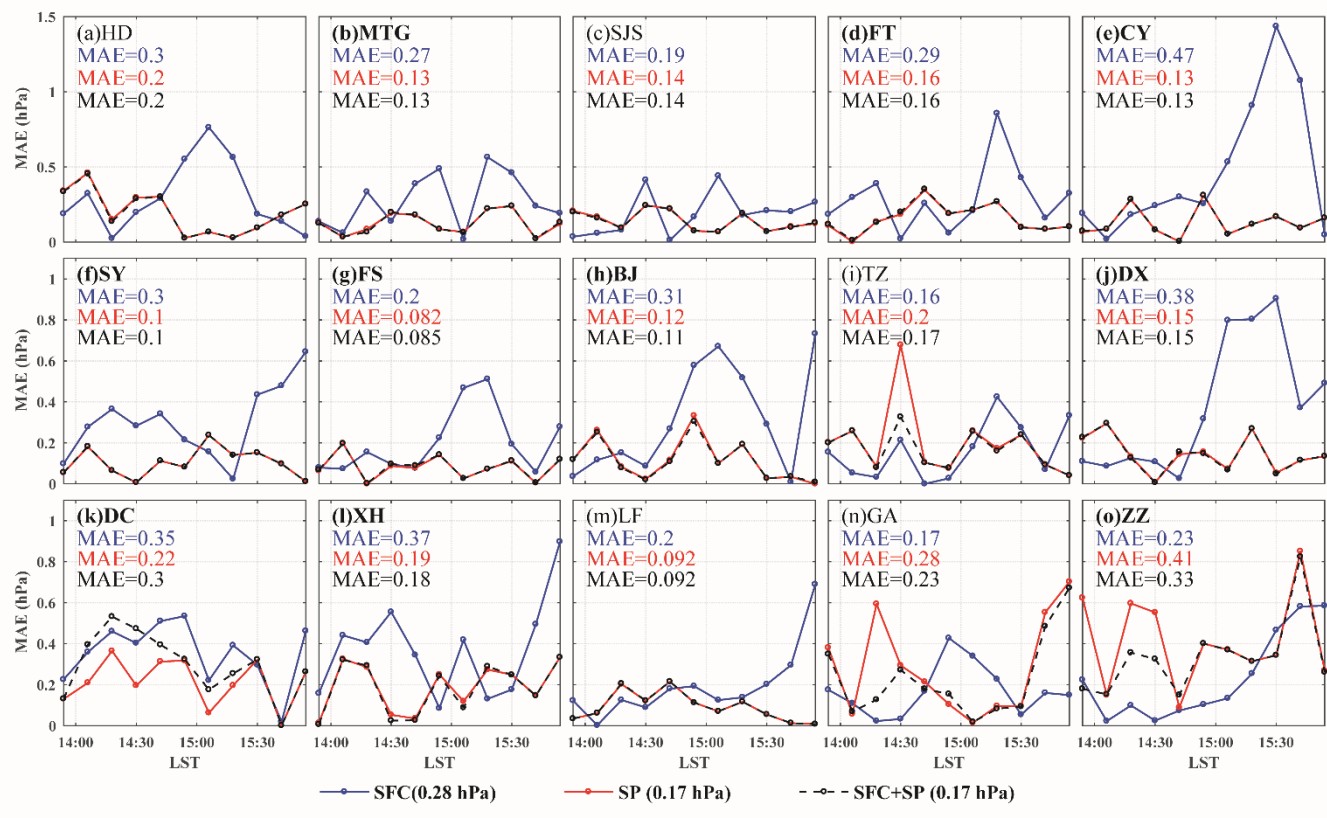

**Figure 12. (a) – (o): Temporal distribution of mean absolute error (MAE) between model analysis and observation at different surface stations for the smartphone experiment (SP; red line), the station observation experiment (SFC; blue line) and the station observation plus smartphone experiment (SFC+SP; black dash line), respectively. The temporally averaged MAE is also shown winthin each plot and the average MAE over all stations for the three experiments are shown below the plots. The bold station names indicate the MAE difference between SFC and SP at those stations is significant with the confidence level of 90%.**





470 **Appendix A**

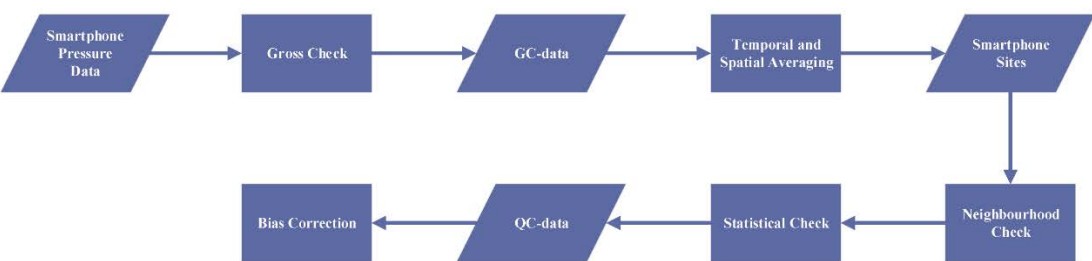

**Figure A1. The workflow for smartphone pressure data quality control and preprocessing. See text for details.**

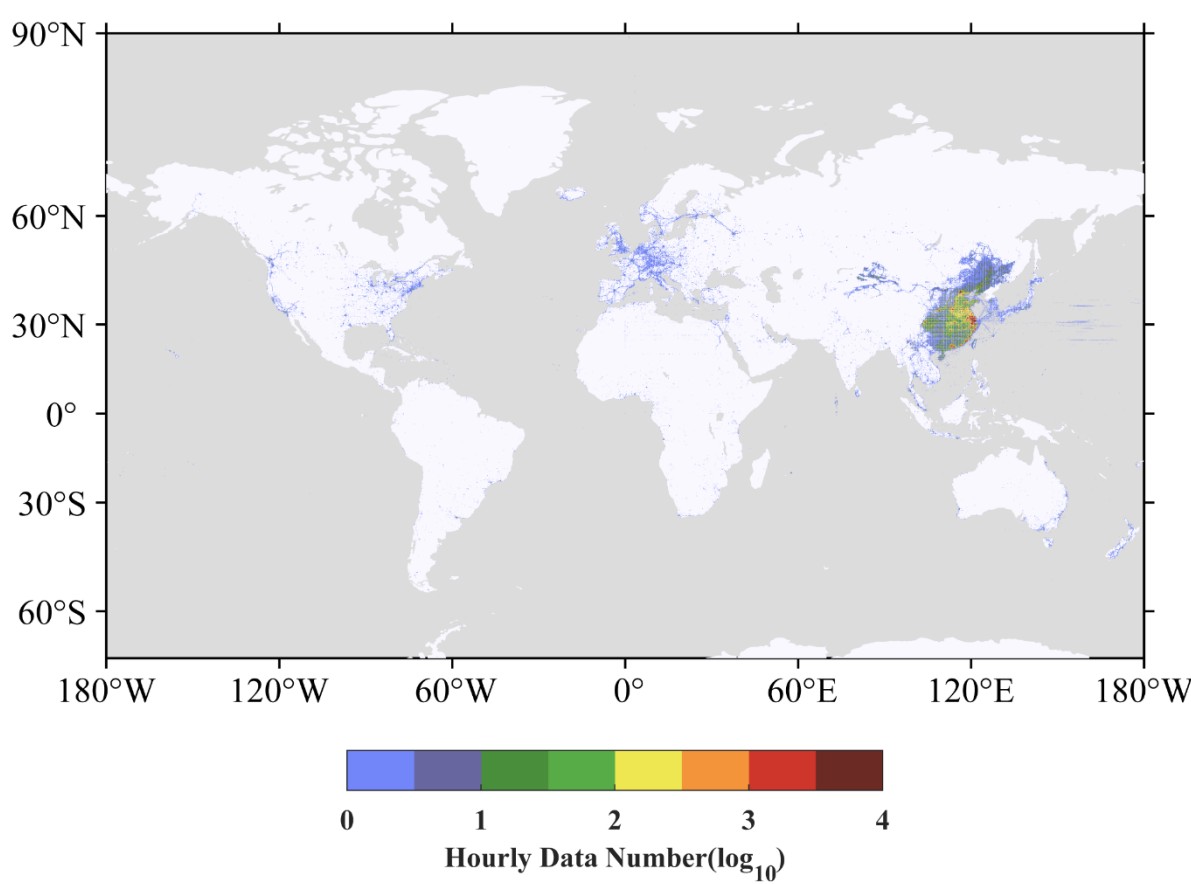

**Figure A2. Hourly pressure observation counts (log10 transformed) averaged over the year 2016. Data are binned into a 0.1° × 0.1°**
475   **grid.**

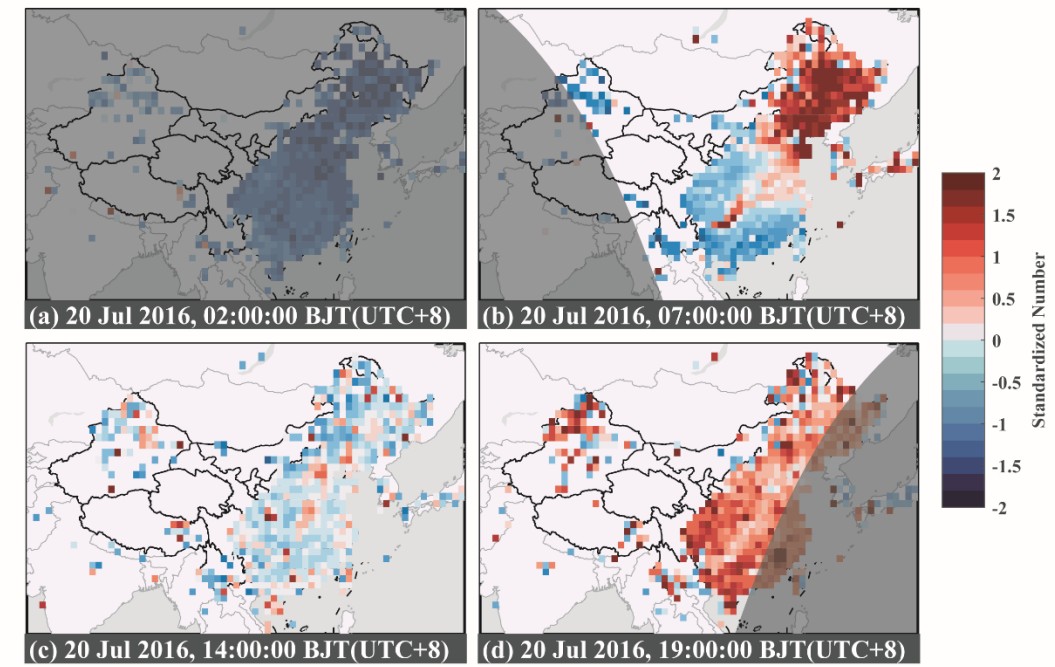

**Figure A3. Spatial distribution of the standardized value of data number at each site for each hour on (a, b) 19 July, 2016, and (c, d) 20 July, 2016. The time is shown in Beijing standard time (BJT). The standardized number is defined as the difference between the data number in this grid at a specific hour and daily mean of the number divided by the standard deviation of the number. The dark gray color fill stands for the region in nighttime. Warm colors indicate a rise in data volume, while cool colors indicate a decrease.**

480

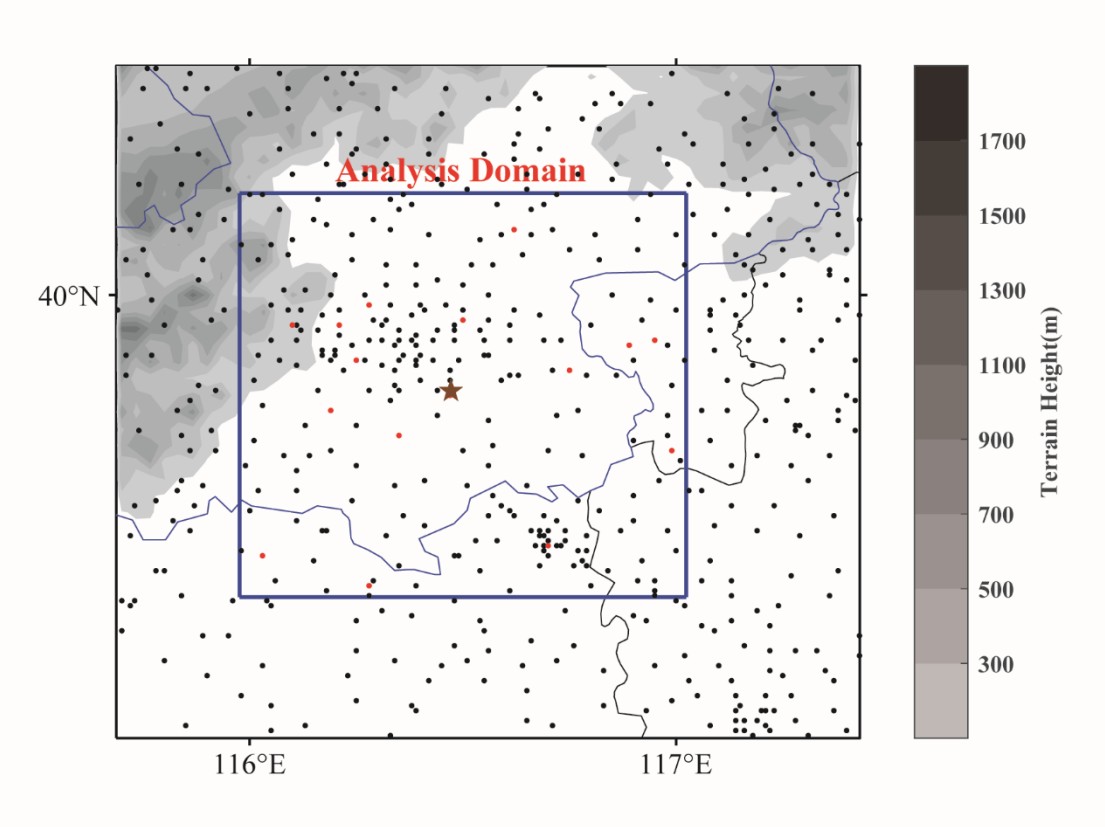

**Figure A4. Objective analysis domain (blue box), terrain height (shaded), the distribution of surface stations (black and red dots with the red dots representing the surface stations with pressure measurements) and Beijing Radar station (star). The boundary of Beijing is shown with blue line.**



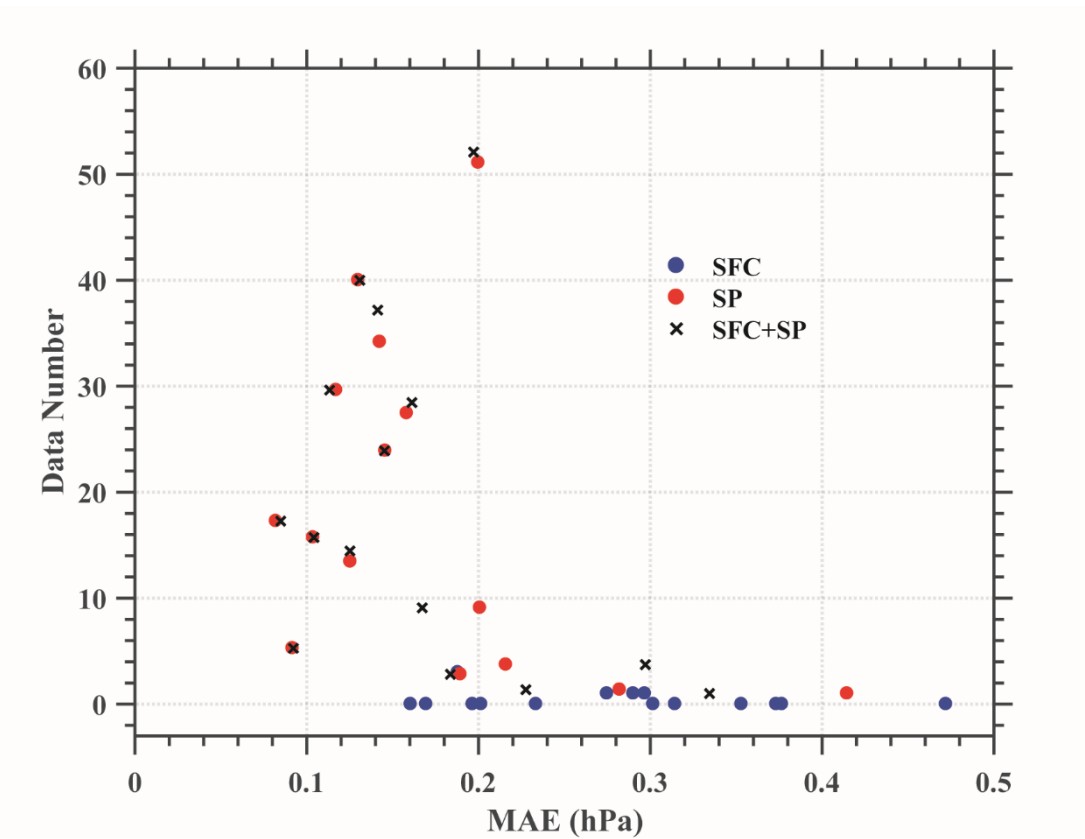

**Figure A5.** Scatter plot of mean absolute error (MAE) versus observation number within 10 kilometers and 5 minutes of the verifying weather station. The blue dots stand for the station observation experiment (SFC), red dots represent the smartphone experiment (SP), and the station observation plus smartphone experiment (SFC+SP) is shown as black crosses.