# Peer review of "Smartphone Pressure Data: Quality Control and Impact on Atmospheric Analysis"

_Atmospheric Measurement Techniques, 2020_

## Referee Comment (RC1) · Colin Price (Referee) · 13 Aug 2020

This is an interesting paper that uses pressure sensors within smartphones to study atmospheric processes. The authors present a methodology of "cleaning" the data and removing the bias, and then present a case study of a hail storm showing the added value of these pressure sensor data from smartphones. The paper is well written, clear, and easy to follow. I only have some minor technical issues and language corrections.

Line 18: hailstorm that occurred...

Line 111: In order to evaluate ....

Line 140-1: what do you mean by two "belts"?

[Figure]

Line 157: How come the App is not open all the time? Cannot it not run in the background?

Line 163-4: How do you see the earlier volume in the northeast China? I do not see this on Fig. A2b. It is not shown any time parameters.

Line 165: hailstorm that occurred.....

Line 217: hence are able....

Line 229-30: Location of radar and surface stations NOT shown in Fig. A3.

Line 231-240: There is a mix up between the text and the figures. The Fig. 10 only has panels for SFC and SCC+SP, and not SP alone. So all letters in text need to be fixed.

Line 235: did you mean Fig. 11? There is no right column in Fig. 10

Line 237: right column?

Line 240: Fig. 11? Please check text against all figures in final version

Line 255 and 259: Fig. 12??

Colin Price Tel Aviv University August 2020

---

## Referee Comment (RC2) · Anonymous Referee #3 · 3 Nov 2020

Review of AMT-2020-190 "Smartphone Pressure Data: Quality Control and Impact on Atmospheric Analysis" By: Rumeng Li, Qinghong Zhang, Juanzhen Sun, Yun Chen,Lili Ding5, and Tian Wang

The manuscript presents an analysis of the global spatial and temporal variation of smartphone pressure measurements using data collected in China by a Weather App every second, in 2016. The Authors present a new bias correction method based on a machine learning approach, showing the potential for the use of this type of dataset in conjunction with surface meteorological-station measurement. The dataset is used to investigate meteorological information during a hailstorm that occurred in Beijing. The most valuable part of this methodology is in the eye of this Referee the fact that the process does not require users' personal information, so avoiding a lot of pushback

for privacy-protection. I think the manuscript is well written, the methodology is clearly presented, and it is a good read. Figures are mostly clear; some suggestions about those are below. The study is well suited for AMT and it will be a good contribution to make better use of this type of information.

Major comments:

Figures: I don't understand the use of numeration A1, A2,... for the figures, since they are just used normally in the text and those identified by the "A" are not part of an Appendix. Would it make sense to just add them to the list of regular figures in the manuscript? In this case Figures have to be renumbered sequentially, as mentioned in the text, and text has to be changed accordingly.

Minor comments:

Page 2, line 57: While in this study you make use of the Moji Weather App, do you envision the same good outcome from other Weather Apps?

Page 4, line 119: Is the threshold of 15 hPa found empirically?

Fig. 2a: Labels "SH" and "TJ" should be a different color (maybe black?) as the one used now makes them not visible. Also, the white circles are not very visible now.

Page 6, lines 159, 160, and 174: Replace "(Fig. 2a.)" with "(Fig. 2a)". Replace "(Fig. 2b.)" with "(Fig. 2b)". Replace "(Fig. A3b.)" with "(Fig. A3b)"

Section 3.2 and Fig. 3: Why do you think in December data count is so low compared to the other months?

Fig 4: Again, the labels "TJ" and "BJ" are not very visible with the chosen color.

Page 7, line 190: Replace "Fig. 5a., b" with "Fig. 5a, b". Similar corrections should be applied in other places in the text (i.e.: Page 8 lines 234, 235; Page 9, line 260; Page 9, line 272...).

[Figure]

Page 7, line 195: Do you mean SMO is not shown in Fig. 6 for subdomain 7 and 8? MP is shown. Also, I would still plot green bars for SMO going to the top of the graphic, for subdomain 7 and 8 (still mentioning in the text that these 2 bars are more than 9 hours long). It would be good to visually see how longer is the computation time for this method.

Page 7, line 212: "(Fig. 7b., d)." change to "(Fig. 7b, d)."

Page 8, line 232: Where does the threshold 0.52 mb come from?

Page 10, line 292: Replace "that we received; and the bias" with "that we received, and the bias"

Fig. 12: To highlight the stations with MAE differences between SFC and SP being significant with 90% confidence try to use underline AND bold.

---

## Author Comment (AC1) · 23 Nov 2020

**Response to Reviewer Comments**
**Smartphone Pressure Data: Quality Control and Impact on Atmospheric Analysis**

Rumeng Li[1], Qinghong Zhang[1*], Juanzhen Sun[2], Yun Chen[3],Lili Ding[4,5], Tian Wang[4]

[1] Department of Atmospheric and Oceanic Sciences, School of Physics, Peking University, Beijing 100871, China

[2] National Center for Atmospheric Science, Boulder, Colorado, United States

[3] National Meteorological Center, Chinese Meteorological Administration, Beijing 100080,China

[4] Moji Co., Ltd, Beijing, 100015, China

[5] Theme Tech Inc, Beijing, 100020, China

*Correspondence to*: Qinghong Zhang (qzhang@pku.edu.cn)

We thank the referees for their time and effort in providing constructive comments, which greatly helped the improvement of the manuscript. Our answers are structured in the following sequence: (1) comments from referees/public (in regular typeset), (2) authors' response (in **bold**), and (3) authors' changes in manuscript (in red and *italic*). Besides, we have numbered the comments from the referees for clarity.

**Response to Referee #1:**

1. This is an interesting paper that uses pressure sensors within smartphones to study atmospheric processes. The authors present a methodology of "cleaning" the data and removing the bias, and then present a case study of a hail storm showing the added value of these pressure sensor data from smartphones. The paper is well written, clear, and easy to follow. I only have some minor technical issues and language corrections.

**Reply:**

**We appreciate the encouraging positive comments from Prof. Price Colin.**

2. Line 18: hailstorm that occurred...

**Reply:**

**Corrected.**

"*The potential application of the high-density smartphone data in cities is illustrated by a case study of a hailstorm that occurred in Beijing in which high-resolution gridded pressure analysis is produced.*"

3. Line 111: In order to evaluate ....

**Reply:**

**We guess that the referee might read the original version of the manuscript because we revised this grammar mistake in our second version manuscript submitted to AMTD, i.e., the revised preprint version. In the preprint version, this sentence, located at Line 120, is changed to:**

"*In order to evaluate the performance of the neighbourhood-based bias correction method,*"

4. Line 140-1: what do you mean by two "belts"?

**Reply:**

**We re-checked the pressure values at these points by comparing them with the nearest ERA5 hourly reanalysis data and found the data in this region were biased, likely due to GPS location error. Hence these data do not appear in the preprint version (Figure R1 below, also Figure 2 in the revised manuscript). And the text has been revised in the main text.**

"*We used the GC-data to analyze the spatial and temporal distribution of the smartphone data counts in 2016. The data location map in Fig. 2 shows that smartphone data are distributed over nearly all continents although most of the data counts occur in China with much higher data density (Fig. 3).*"

[Figure]

**Figure R1. Locations of global pressure observations in 2016 from the Moji Weather application.**

5. How come the App is not open all the time? Cannot it not run in the background?

**Reply:**

**The app can receive data from users only when 1) the network is available and 2) the app is open on either the front end or the back end. So, it can run in background but most users tend to close the background application when not in use to save power. When the network is unavailable, the app cannot receive the pressure value. The text was revised in the manuscript.**

*"owing to the fact that the app can only get the pressure information when the network is available and when the users open the app either on the front end or on the back end."*

6. Line 163-4: How do you see the earlier volume in the northeast China? I do not see this on Fig. A2b. It is not shown any time parameters.

**Reply:**

**From Fig. 6 in the revised manuscript (Fig. R2 attached below), when it is early daytime (0700 BJT), the data volume is increasing in the northeast China (the**

**warm color region), but in south of China, the data volume hasn't started to increase (the cold color region).**

[Figure]

**Figure R2. Spatial distribution of the standardized value of data number at each site for each hour on (a, b) 19 July, 2016, and (c, d) 20 July, 2016.    The time is shown in Beijing standard time (BJT). The standardized number is defined as the difference between the data number in this grid at a specific hour and daily mean of the number divided by the standard deviation of the number. The dark gray color fill stands for the region in nighttime. Warm colors indicate a rise in data volume, while cool colors indicate a decrease.**

7. Line 165: hailstorm that occurred.....

**Reply:**

**Corrected.**

"*Analysis during a hailstorm that occurred in Beijing further reveals that people respond promptly to severe weather event.*"

8. Line 217: hence are able....

**Reply:**

**Corrected.**

  "*the smartphone pressure observations are much denser and hence are able to capture the fine-scale pressure change associated with the cold pool*"

9. Line 229-30: Location of radar and surface stations NOT shown in Fig. A3.

**Reply:**

**We added the location of radar and surface stations in Fig. 13 in the revised manuscript (Fig. R3 attached below).**

[Figure]

**Figure R3. Objective analysis domain (blue box), terrain height (shaded), the distribution of surface stations (black and red dots with the red dots representing the surface stations with pressure measurements) and Beijing Radar station (star). The boundary of Beijing is shown with blue line.**

10. Line 231-240: There is a mix up between the text and the figures. The Fig. 10 only has panels for SFC and SCC+SP, and not SP alone. So all letters in text need to be fixed.

**Reply:**

**Fig. 10 and Fig. 11 are changed into Fig. 14 and Fig. 15 (Fig. R4 and Fig. R5 attached below) in the revised manuscript and the text has also been changed accordingly.**

[Figure]

**Figure R4. Surface pressure perturbation analyses (shaded) from the experiment SFC (a, c, e) and SFC+SP (b, d, f), overlaid by VDRAS wind field at 150 m (thick black arrows) and column maximum radar reflectivity (contours). The valid time is 1500 LST for the top row, 1506 LST for the middle row, and 1512 LST for the bottom row.**

[Figure]

**Figure R5. Vertical cross section of the radar reflectivity (shaded), VDRAS wind field (thin black arrows) and vertical velocity field (brown contours with dash lines for downward motion and solid lines for upward motion) along the line A-B in the Figure 10. The solid blue line and red dash line are the surface pressure perturbation along the A-B line from SFC and SFC+SP, respectively.**

11. Line 235: did you mean Fig. 11? There is no right column in Fig. 10

**Reply:**

**The vertical cross sections are shown in Fig. 15 in the revised manuscript (Fig. R5 above).**

"*The vertical cross sections shown in Fig. 15 through the line A-B (see Fig. 14) indicate that*"

12. Line 237: right column?

**Reply:**

**Yes, it's the right column and this sentence is revised in the revised manuscript.**

"*We first note that the perturbation pressure analysis from SFC+SP (right column) displays small-scale features in and around the storm that are absent in SFC (left column).*"

13. Line 240: Fig. 11? Please check text against all figures in final version

**Reply:**

**Thanks for the suggestion, yes, it is now shown in Fig. 15. This sentence has been changed in the revised manuscript. We also checked the text and figures again to make sure they are consistent.**

"*The vertical cross sections shown in Fig. 15 through the line A-B (see Fig. 14) indicate that*"

14. Line 255 and 259: Fig. 12??

**Reply:**

**Fig. 12 has been changed into Fig. 16 in the revised manuscript.**

**Response to Referee #2:**

1. The manuscript presents an analysis of the global spatial and temporal variation of smartphone pressure measurements using data collected in China by a Weather App every second, in 2016. The Authors present a new bias correction method based on a machine learning approach, showing the potential for the use of this type of dataset in conjunction with surface meteorological-station measurement. The dataset is used to investigate meteorological information during a hailstorm that occurred in Beijing. The most valuable part of this methodology is in the eye of this Referee the fact that the process does not require users' personal information, so avoiding a lot of pushback for privacy-protection. I think the manuscript is well written, the

methodology is clearly presented, and it is a good read. Figures are mostly clear; some suggestions about those are below. The study is well suited for AMT and it will be a good contribution to make better use of this type of information.

**Reply:**

**We thank the reviewer for the positive evaluation for our manuscript.**

2. Figures: I don't understand the use of numeration A1, A2,: : : for the figures, since they are just used normally in the text and those identified by the "A" are not part of an Appendix. Would it make sense to just add them to the list of regular figures in the manuscript? In this case Figures have to be renumbered sequentially, as mentioned in the text, and text has to be changed accordingly.

**Reply:**

**Thanks for the suggestion, we have added those figures to the list of regular figures and changed the main text accordingly.**

3. Page 2, line 57: While in this study you make use of the Moji Weather App, do you envision the same good outcome from other Weather Apps?

**Reply:**

**Yes, we think for other Weather Apps they can also have the same good results as they work in the similar manner except that the data count may be different from different user groups.**

4. Page 4, line 119: Is the threshold of 15 hPa found empirically?

**Reply:**

**We calculated the percentage of effective data for different thresholds, i.e., the data count of eliminated data after a certain threshold is applied divided by the data count of original data, and found out that the threshold of 15 hPa can retain relatively large amounts of original data. Besides, the standard deviation of the departure of smartphone data and automatic weather station is about 2.59 hPa, so 15 hPa is about six times the standard deviation.**

[Figure]

**Figure R6. Distribution of the percentage of effective data count at different pressure threshold.**

5. Fig. 2a: Labels "SH" and "TJ" should be a different color (maybe black?) as the one used now makes them not visible. Also, the white circles are not very visible now.

**Reply:**

**Labels "SH" and "TJ" have been changed into black colour and larger size. And the colour of circles have also been changed into red. The text was revised in the manuscript.**

*"such as the densely populated urban agglomerations of the Yangtze River Delta (Shanghai and nearby cities), Pearl River Delta (Guangzhou and nearby cities), and Beijing-Tianjin-Hebei region (marked by red circles in Fig. 4a)"*

**The new figure is as follow.**

[Figure]

**Figure R7. (a) Hourly pressure observation counts (log10 transformed) averaged over the year 2016. (b) Same as a but for the China Meteorological Administration (CMA) surface stations. Data are binned into a 0.1° × 0.1° grid in (a) - (b). The location of the port of Shanghai and the port of Tianjin are labelled as "SH" and "TJ" in (a). The red circles indicate the urban agglomerations of (from north to south) Beijing-Tianjin-Hebei region, Shanghai and nearby cities, and Guangzhou and nearby cities.**

6. Page 6, lines 159, 160, and 174: Replace "(Fig. 2a.)" with "(Fig. 2a)". Replace "(Fig. 2b.)" with "(Fig. 2b)". Replace "(Fig. A3b.)" with "(Fig. A3b)"

**Reply:**

**Corrected.**

7. Section 3.2 and Fig. 3: Why do you think in December data count is so low compared to the other months?

**Reply:**

The data count in December is below $2 \times 10^5$ month$^{-1}$h$^{-1}$ while for other months, data counts are above $2 \times 10^5$ month$^{-1}$h$^{-1}$. And we think the reason may be that the app can receive data from users only when the network is available and the app is open, so the data count is largely influenced by the users' behaviour. People are more likely to open the app and check the weather when severe weathers happen. In winter, there are less severe weathers and people do not open the app as frequently as in summer, so the data count is less.

8. Fig 4: Again, the labels "TJ" and "BJ" are not very visible with the chosen color.

**Reply:**

**In order to give a better view, the labels have been enlarged and changed into yellow. The new figure is as follow.**

[Figure]

**Figure R8. Diurnal variation of the data volume for smartphone data on the day of the hailstorm (red line) and the annual mean value (blue line) for 39°N–41°N, 115°E–118°E. (a)–(d) show a 3D view of data counts in a 0.05° × 0.05° grid over 6 minutes (colored columns) before each radar volume and radar echo (grey columns). The color and height of each column represent the value of the data count. BJ, Beijing; TJ, Tianjin.**

9. Page 7, line 190: Replace "Fig. 5a., b" with "Fig. 5a, b". Similar corrections should be applied in other places in the text (i.e.: Page 8 lines 234, 235; Page 9, line 260; Page 9, line 272...).

**Reply:**

**Thanks for the careful reading. They have been revised.**

10. Page 7, line 195: Do you mean SMO is not shown in Fig. 6 for subdomain 7 and 8? MP is shown. Also, I would still plot green bars for SMO going to the top of the graphic, for subdomain 7 and 8 (still mentioning in the text that these 2 bars are more than 9 hours long). It would be good to visually see how longer is the computation time for this method.

**Reply:**

**Thanks for the correction and suggestion, it is for SMO and sorry for the mismatch of (a) and (b), the subdomain indexes in (b) is not right, it should be for subdomain 2 and 6, where the computation time is more than 9 hours, we have corrected it. The revised picture is shown below. And the text has also been revised in the manuscript.**

*"The computation times for subdomain 2 and subdomain 6 using SMO are more than 9 hours."*

[Figure]

**Figure R9. (a) Mean absolute error (MAE) distribution at different training subdomains for different machine learning methods (MP refers to Multilayer Perception method; SMO refers to Support Vector Machine method; RF refers to Random Forest method). (b) is the same as (a), but for computation time.**

11. Page 7, line 212: "(Fig. 7b., d)." change to "(Fig. 7b, d)."

**Reply:**

**Corrected.**

*"by 16% for the additional sites (Fig. 10b, d)"*

12. Page 8, line 232: Where does the threshold 0.52 mb come from?

**Reply:**

**We tried different thresholds, for 1536 LST, the threshold of 0.52 mb can give the best estimation of the cold pool leading edge; for other times, different thresholds didn't have too much difference.**

13. Page 10, line 292: Replace "that we received; and the bias" with "that we received, and the bias"

**Reply:**

**Corrected.**

*"no private information was included in the raw data that we received, and the bias correction method proposed"*

14. Fig. 12: To highlight the stations with MAE differences between SFC and SP being significant with 90% confidence try to use underline AND bold.

**Reply:**

**Thanks for the suggestion. We have highlighted the stations with significant difference using underline and bold. The revised figure is as follow.**

[Figure]

**Figure R10. (a) – (o): Temporal distribution of mean absolute error (MAE) between model analysis and observation at different surface stations for the smartphone experiment (SP; red line), the station observation experiment (SFC; blue line) and the station observation plus smartphone experiment (SFC+SP; black dash line), respectively. The temporally averaged MAE is also shown winthin each plot and the average MAE over all stations for the three experiments are shown below the plots. The underlined and bold station names indicate the MAE difference between SFC and SP at those stations is significant with the confidence level of 90%.**